

# Transcriptome for the breast muscle of Jinghai yellow chicken at early growth stages

Pengfei Wu, Xinchao Zhang, Genxi Zhang, Fuxiang Chen, Mingliang He, Tao Zhang, Jinyu Wang, Kaizhou Xie and Guojun Dai

College of Animal Science and Technology, Yangzhou University, Yangzhou, China
Joint International Research Laboratory of Agriculture & Agri-Product Safety, Yangzhou University, Yangzhou, China

Corresponding author
Genxi Zhang, gxzhang@yzu.edu.cn

## ABSTRACT

**Background.** The meat quality of yellow feathered broilers is better than the quality of its production. Growth traits are important in the broiler industry. The exploration of regulation mechanisms for the skeletal muscle would help to increase the growth performance of chickens. At present, some progress has been made by researchers, but the molecular mechanisms of the skeletal muscle still remain unclear and need to be improved.

**Methods.** In this study, the breast muscles of fast- and slow-growing female Jinghai yellow chickens (F4F, F8F, F4S, F8S) and slow-growing male Jinghai yellow chickens (M4S, M8S) aged four and eight weeks were selected for transcriptome sequencing (RNA-seq). All analyses of differentially expressed genes (DEGs) and functional enrichment were performed. Finally, we selected nine DEGs to verify the accuracy of the sequencing by qPCR.

**Results.** The differential gene expression analysis resulted in 364, 219 and 111 DEGs (adjusted $P$-value $\leq$ 0.05) for the three comparison groups, F8FvsF4F, F8SvsF4S, and M8SvsM4S, respectively. Three common DEGs (ADAMTS20, ARHGAP19, and Novel00254) were found, and they were all highly expressed at four weeks of age. In addition, some other genes related to growth and development, such as ANXA1, COL1A1, MYH15, TGFB3 and ACTC1, were obtained. The most common DEGs ($n = 58$) were found between the two comparison groups F8FvsF4F and F8SvsF4S, and they might play important roles in the growth of female chickens. The Kyoto Encyclopedia of Genes and Genomes (KEGG) pathway also showed some significant enrichment pathways, for instance, extracellular matrix (ECM)-receptor interaction, focal adhesion, cell cycle, and DNA replication. The two pathways that were significantly enriched in the F8FvsF4F group were all contained in that of F8SvsF4S. The same two pathways were ECM–receptor interaction and focal adhesion, and they had great influence on the growth of chickens. However, many differences existed between male and female chickens in regards to common DEGs and KEGG pathways. The results would help to reveal the regulation mechanism of the growth and development of chickens and serve as a guideline to propose an experimental design on gene function with the DEGs and pathways.

## INTRODUCTION

Chicken contains a variety of nutrients and is also suitable for different kinds of people (*Zhang et al., 2019a*). Accordingly, chicken has been well received and is the second largest meat product in China. However, with the improvement of people's living standards, the demand for chicken is changing from "quantity" to "quality". Yellow feathered broilers just meet this demand of the people in China because of the better meat quality and disease resistance. However, the growth rate, muscle yield, and feed efficiency of yellow feathered broilers are all inadequate compared with white feathered broilers. Therefore, these performances still have great potential to be improved in yellow feathered broilers.

Growth traits are important economic traits in broiler production. Increasing the growth rate of broiler chickens has always been one of the goals pursued by breeders (*Papah et al., 2018*), and the growth rate has increased considerably in the past few decades (*Fontana et al., 2017*). The growth of broilers is generally influenced by environment and heredity, and nowadays, under the condition of industrialized production, the most important factor affecting the growth of broilers is heredity. Growth trait is a quantitative trait regulated by multiple genes (*Goto et al., 2019*), such as myogenic regulatory factors (MRFs) (*Zhu et al., 2016*), insulin like growth factor 1/insulin like growth factor 1 receptor (IGF-I/IGF-IR) (*Roelfsema et al., 2018*), myostatin (MSTN) (*Liu et al., 2016*), etc. Therefore, it is necessary to study the molecular mechanism of growth regulation in poultry.

With the advent of the post-genomic era, transcriptomics, proteomics, metabolomics, and other omics technologies have emerged one after another (*Prucha, Zazula & Russwurm, 2018*). The transcriptomic technology has developed rapidly and has been increasingly used in recent years. Transcriptome sequencing (RNA-seq) provides a measurement of levels of transcripts for biological samples with the next-generation sequencing technology (*Wang, Gerstein & Snyder, 2009*). In recent years, RNA-seq has been widely used in various fields of poultry and has become an important tool for researchers to explore the regulation mechanism of growth and development of poultry.

*Zhang et al. (2019a)* carried out a transcriptome study on the breast muscles of three chicken strains (White Broiler, Daheng, and Commercial Layers of Roman) and a total of 8398 DEGs were obtained. They found some DEGs related to muscle growth, including MYH15, MYOZ2, MYBPC3, IGF2, BCL-2, JUN, and FOS. Kyoto Encyclopedia of Genes and Genomes (KEGG) pathway enrichment analysis showed that extracellular matrix (ECM)–receptor interaction, the mitogen-activated protein kinase (MAPK) signaling pathway, and focal adhesion were the most enriched for the DEGs. In order to study the effect of intramuscular preadipocytes (IMPA) on muscle development, primary skeletal muscle satellite cells (MSC) and IMPA were isolated from the pectoralis major muscle of seven-day-old chickens by *Guo et al. (2018)*. MSC were cultured alone or co-cultured with IMPA for 2 d and then subjected to RNA-seq. The results showed that most of DEGs related to muscle development were downregulated in co-cultured MSC, and DEGs related to lipid deposition were upregulated. Pathway analysis indicated that IMPA might inhibit a differentiation via the JNK/MAPK pathway and promote lipid deposition via the PPAR pathway. *Ren et al. (2018)* collected six-week-old pectoral muscles of slow-growing (Gushi,

GS) and fast-growing (Arbor Acres, AA) chicken breeds for transcriptome sequencing. A total of 4815 differentially expressed lncRNAs (long non-coding RNAs) were screened. Finally, two lncRNAs specifically expressed in muscle tissues, the TCONS_00064133 and the TCONS_00069348. Although RNA-seq technology has been applied to study the growth and development of poultry, the specific regulatory mechanisms of skeletal muscle development remain unclear, and the transcriptome sequencing technology still will be increasingly explored in future studies in the field.

In chickens, breast muscle is a major contributor to the skeletal muscle and is directly correlated with meat quantity and quality. Therefore, exploring the molecular mechanisms underlying skeletal muscle development has been a focus of research in the field of poultry genetic breeding (*Li et al., 2019*). In this study, the breast muscles of Jinghai yellow chicken were collected for RNA-seq. We expect to find genes and pathways related to growth and development of Jinghai yellow chickens. The results will provide a theoretical basis for the breeding of Jinghai yellow chicken and will also contribute to the further improvement of the growth and development regulation mechanism of chickens.

## MATERIALS & METHODS

### Ethics statement

The animal experiments performed in the study were all evaluated and approved by the Animal Ethics Committee of Yangzhou University (Yzu DWLL-201903-001).

### Experimental animals and sample collection

The Jinghai yellow chickens used in the study were obtained from Jiangsu Jinghai Poultry Industry Group Co., Ltd. (Nantong City, Jiangsu Province, China). This chicken breed is also the first female parent of a national chicken breed, Haiyang yellow chicken, which was approved by National Livestock and Poultry Genetic Resources Committee in 2018. The fast-growth and the slow-growth groups of Jinghai yellow chickens were hatched on the same day and raised separately on the floor in the same chicken house until transferring them to laying cages at 14 weeks of age, where they had access to feed and water ad libitum. At the ages of 4 and 8 weeks, we selected three healthy individuals with similar body weight from the slow-growing male chickens (M4S and M8S), slow-growing female chickens (F4S and F8S), and fast-growing female chickens (F4F and F8F), respectively. All chickens were euthanized by carotid artery bloodletting after being anesthetized by intravenous injection of 8 mg/kg of xylazine hydrochloride (SIGMA, Japan). The left breast muscles were then collected and stored at −80 °C for RNA extraction and RNA-seq.

### The construction of a cDNA library and sequencing

Total RNA from the breast muscles was isolated after a month using the TRIzol total RNA Extraction Kit (Invitrogen, Carlsbad, CA, USA), according to the manufacturer's instructions. A total amount of 3 µg RNA per sample was used as input material for the RNA sample preparations. Sequencing libraries were generated using the NEBNext® Ultra™ RNA Library Prep Kit for Illumina® (NEB), following the manufacturer's recommendations, and index codes were added to attribute sequences to each sample. The

library preparations were sequenced on the Illumina NovaSeq 5000 platform and 150 bp paired-end reads were generated.

## Quality control

Raw data (raw reads) of fastq format were firstly processed through in-house perl scripts. In this step, clean data (clean reads) were obtained by removing reads containing adapter, reads containing ploy-N, and low quality reads from raw data. At the same time, Q20, Q30, and GC content of the clean data were calculated. All the downstream analyses were based on the clean data with high quality.

## Reads mapping and quantification of gene expression level

We selected the HISAT (*Kim, Langmead & Salzberg, 2015*) software to align the filtered sequence to the chicken genome. HTSeqv0.6.1 (*Anders, Pyl & Huber, 2015*) was used to count the reads numbers mapped to each gene. Then, the expected number of fragments per kilobase of transcript sequence per million base pairs sequenced (FPKM) of each gene was calculated based on the length of the gene and reads count mapped to this gene. FPKM, expected number of fragments per kilobase of transcript sequence per million base pairs sequenced, considers the effect of sequencing depth and gene length for the reads count at the same time, and is currently the most commonly method used for estimating gene expression levels (*Trapnell et al., 2010*).

## Screening and functional analysis of DEGs

Differential expression analysis was performed using the DESeq R package (1.18.0) (*Anders & Huber, 2010*). DESeq provides statistical routines for determining differential expression in digital gene expression data using a model based on the negative binomial distribution. The resulting *P*-values were adjusted using the Benjamini and Hochberg's (BH) approach for controlling the false discovery rate. Genes with an adjusted *P*-value ≤0.05 found by DESeq were assigned as differentially expressed.

Gene Ontology (GO) enrichment analysis of DEGs was implemented by the GOseq R package (*Young et al., 2010*) in which gene length bias was corrected. GO terms with a corrected *P*-value ≤0.05 were considered significantly enriched by differential expressed genes. We used the KOBAS (*Mao et al., 2005*) software to test the statistical enrichment of differential expression genes in KEGG pathways. KEGG (Kyoto Encyclopedia of Genes and Genomes) pathways with a corrected *P*-value ≤0.05 were considered significantly enriched. KEGG is a database resource for understanding high-level functions and utilities of the biological system, such as the cell, the organism, and the ecosystem, from molecular-level information, especially large-scale molecular datasets generated by genome sequencing and other high-throughput experimental technologies (http://www.genome.jp/kegg/).

## Validation of DEGs by qPCR

RNA samples used in qPCR was the same as RNA-seq. Reverse transcription of mRNA to cDNA was operated according to the instructions of the HiScript Q Select RT SuperMix for qPCR kit (R123-01, Vazyme, Nanjing, China). The primers used for quantification in the study were designed using Primer 5.0, and β-actin was used as housekeeping gene. The
**Table 1    Average weight of Jinghai yellow chickens at 4 and 8 weeks used in the study.**

| Sample group | 4-week-old weight (g) | 8-week-old weight (g) |
| --- | --- | --- |
| Fast-growing female chicken(FF) | $306.67 \pm 5.77^A$ | $761.67 \pm 16.07^B$ |
| Slow-growing female chicken(FS) | $203.33 \pm 22.55^A$ | $540.00 \pm 35.00^B$ |
| Slow-growing male chicken(MS) | $210.00 \pm 32.79^A$ | $571.67 \pm 40.41^B$ |

**Notes.**
The difference in the same row with different capital letters was considered as significant ($P < 0.01$).

sequences of primers are shown in Table S1. The qPCR was conducted on the Applied Biosystems 7500 real-time PCR system (Applied Biosystems) using the ChamQ SYBR qPCR Master Mix kit (Q311-02; Vazyme, Nanjing, China). The relative expression of genes was calculated using the $2^{-\Delta\Delta CT}$ method.

# RESULTS

## Comparison of body weight between different weeks of age

Significant differences in body weight between different weeks of age were measured by independent sample $T$-tests in SPSS 13.0. The results (Table 1) showed that there were significant differences among the three comparison groups at different weeks of age ($P < 0.01$), illustrating that the growth rate of Jinghai yellow chickens was faster during a four-week period.

## Quality control of sequencing data

The results of quality assessment of sequencing data are detailed in Table S2. Clean bases of each sample reached 7.17 G (M4S_3) or more, which meets the requirement of sequencing data quantity. The percentage of the clean base with 99% correct recognition rate in sample F4F_1 is the lowest (95.55%). The GC content in each sample ranged from 50.82% to 54.80%, indicating that there was no base separation. The sequencing data is good and can be used for a series of subsequent data analysis. All the above results showed that clean data quality is high and it could be used for downstream analyses.

## Reads mapping to the reference genome

We compared the high-quality clean reads obtained from raw data with the reference genome of chickens and the results are shown in Table S3. In each sample, the clean reads mapped to the reference genome were all over 73.88% (M4S_3), which exceeded the standard of 70%. The highest percentage of clean reads mapped to multiple loci in the genome was 4.37% (F4F_1), which did not exceed the limit of 10%. In addition, comparisons and statistical analysis showed that the proportions of uniquely mapped, reads map to '+', reads map to '−', non-splice reads, and splice reads were all within the normal range.

## Analysis of differentially expressed genes

Figure 1 shows the DEG with adjusted $P$-values of ≤0.05. The analysis resulted in 364, 219, and 111 DEGs obtained from the three comparison groups F8FvsF4F (Fig. 1A ), F8SvsF4S

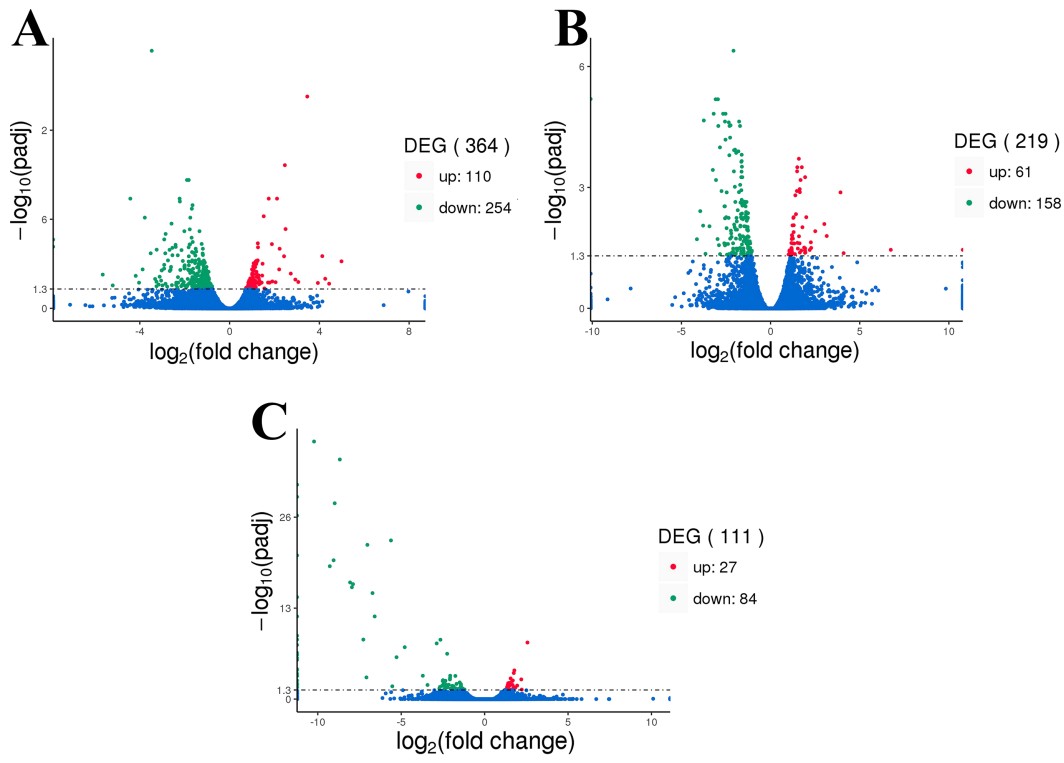

**Figure 1** **The volcano plot of DEGs.** (A) Differentially expressed genes of F8FvsF4F. (B) Differentially expressed genes of F8SvsF4S. (C) Differentially expressed genes of M8SvsM4S.

(Fig. 1B), and M8SvsM4S (Fig. 1C), respectively. Compared to the F4F, the F8F presented 110 upregulated genes and 254 downregulated genes. There were 61 upregulated genes and 158 downregulated genes in comparison of F8S and F4S. And, 27 upregulated genes and 84 downregulated genes were found in the M8SvsM4S group.

The Venn plot with the three comparison groups is shown in Fig. 2. Fifty and eight DEGs were found in F8FvsF4F and F8SvsF4S comparisons, respectively, which indicated that these genes might play important roles in the development of fast-growing and slow-growing Jinghai yellow female chickens. Eighteen DEGs were found in F8SvsF4S and M8SvsM4S. These genes might play important roles in the growth stage of both slow-growing male and female Jinghai yellow chickens. Only seven DEGs were found in both F8FvsF4F and M8SvsM4S groups. The common DEGs in the three comparison groups were ADAMTS20 (ADAM metallopeptidase with thrombospondin type 1 motif 20), ARHGAP19 (Rho GTPase activating protein 19), and Novel00254. These three genes were significantly expressed in both different groups and sexes of Jinghai yellow chickens.

In order to verify the accuracy of biological duplication, we conducted cluster analysis of DEGs in each group. The clustering results are shown in Figs. 3–5, demonstrating that the three individuals in each group were all well clustered together. Also, these results show that the three individuals selected in each group have good repeatability.
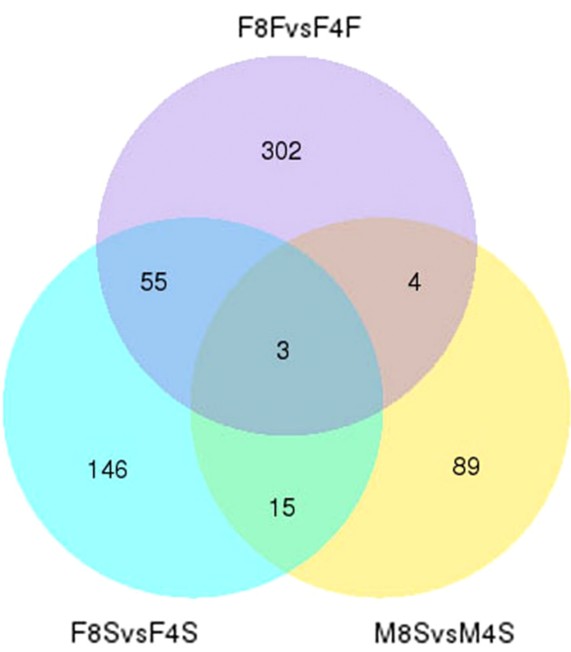

**Figure 2** **The venn plot of DEGs.**

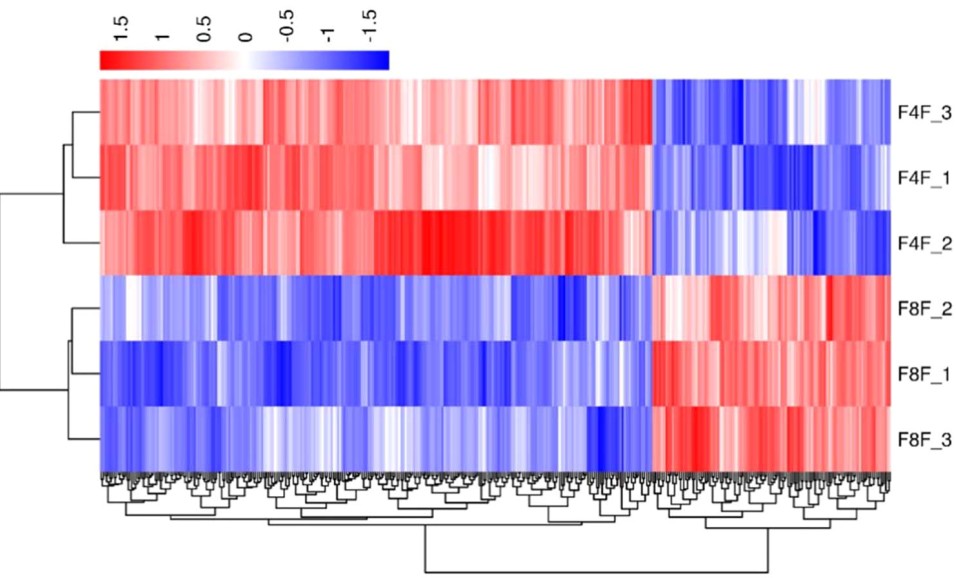

**Figure 3** **The cluster for DEGs of F8FvsF4F.** Different rows represent different samples and different columns represent different genes. Different colors represent the level of gene expression for the samples; red indicates high level expression of genes, while the blue indicates low level expression of genes.

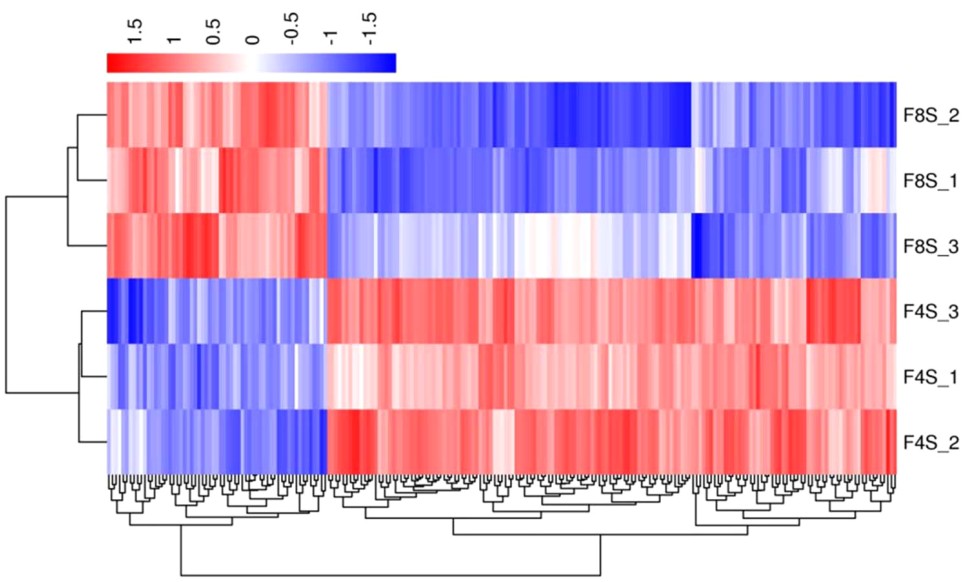

**Figure 4 The cluster for DEGs of F8SvsF4S.** Different rows represent different samples and different columns represent different genes. Different colors represent the level of gene expression for the samples; red indicates high level expression of genes, while the blue indicates low level expression of genes.

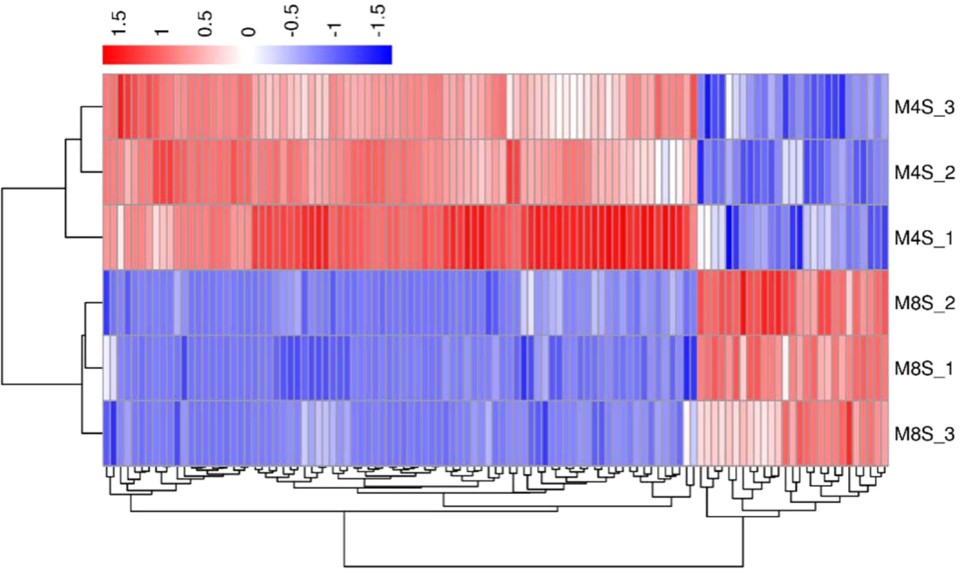

**Figure 5 The cluster for DEGs of M8SvsM4S.** Different rows represent different samples and different columns represent different genes. Different colors represent the level of gene expression for the samples; red indicates high level expression of genes, while the blue indicates low level expression of genes.

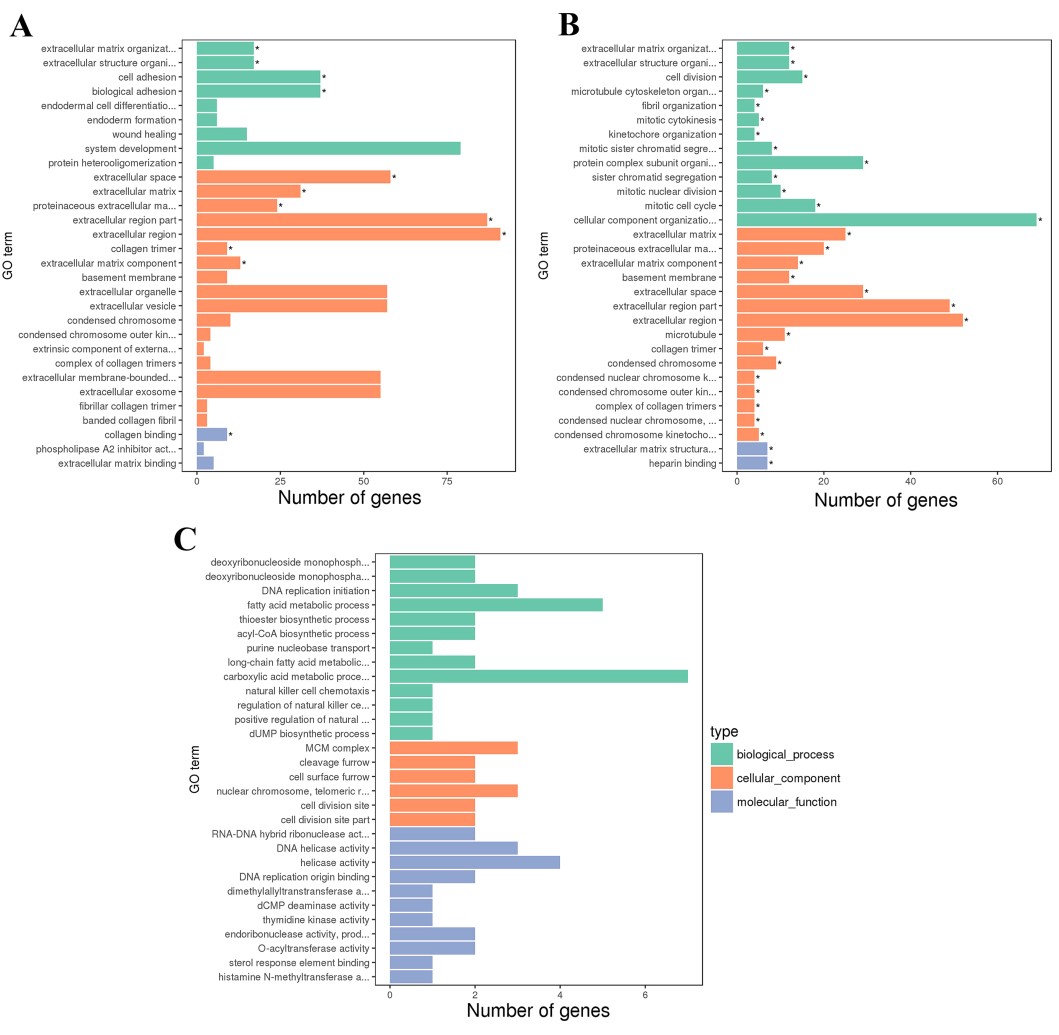

**Figure 6   The GO terms of DEGs.** (A) The first 30 GO terms of DEGs for F8FvsF4F. (B) The first 30 GO terms of DEGs for F8SvsF4S. (C) The first 30 GO terms of DEGs for M8SvsM4S.

## GO and KEGG pathway enrichment analysis

The GO enrichment analysis of DEGs in each group was performed, and the top 30 GO terms of the three comparison groups are shown in Fig. 6. Significantly enriched GO terms were not found in the M8SvsM4S (Fig. 6C) group. For the F8FvsF4F (Fig. 6A) and F8SvsF4S (Fig. 6B) groups, 12 and 44 GO terms were significantly enriched, within these 4 and 17 were classified as biological processes (BP), respectively. In Table 2, we highlighted some important BP terms related to growth and development, such as extracellular matrix organization, extracellular structure organization, cell adhesion, cell division, and fibril organization.

The KEGG pathway analysis of DEGs in the three comparison groups was performed, and the top 20 KEGG pathways of each group are shown in Fig. 7. Most of the first 20 KEGG pathways in each group were related to growth and development, such as regulation of actin

**Table 2  The BP terms related to the growth and development of skeletal muscle in Jinghai yellow chicken.**

| Group | GO accession | BP term | Corrected $p$-Value | DEGs | Important genes |
|---|---|---|---|---|---|
| F8FvsF4F | GO:0030198 | extracellular matrix organization | 0.00025248 | 17 (7) | COL5A1, ANXA2, KAZALD1, ITGA8, PDGFRA, LCP1, TGFBI |
| | GO:0043062 | extracellular structure organization | 0.00025248 | 17 (7) | COL5A1, ANXA2, ITGA8, LCP1, POSTN, TGFBI, COL5A2 |
| | GO:0007155 | cell adhesion | 0.013015 | 37 (10) | COL5A1, STK10, TGFBR3, CYTH1, ITGA8, MYOC, TGFBI, SDK2, DPP4, ZNF703 |
| | GO:0022610 | biological adhesion | 0.013015 | 37 (11) | COL5A1, TLN1, TGFBR3, CNTN1, ANXA1, POSTN, MYOC, TGFBI, SDK2, ZNF703, COL6A2 |
| F8SvsF4S | GO:0030198 | extracellular matrix organization | 0.0013443 | 12 (7) | COL5A1, LAMC1, HSPA8, LTBP2, PXDN, COL5A2, COL3A1 |
| | GO:0043062 | extracellular structure organization | 0.0013443 | 12 (7) | COL5A1, HSPA8, LTBP2, COL5A2, COL3A1, COL18A1, PXDN |
| | GO:0051301 | cell division | 0.0036042 | 15 (8) | TOP2A, PLK1, KIF23, ECT2, CDT1, KIF18B, ASPM, RACGAP1 |
| | GO:1902850 | microtubule cytoskeleton organization involved in mitosis | 0.019561 | 6 (6) | PLK1, KIF23, CDC20, CDT1, NDC80, RACGAP1 |
| | GO:0097435 | fibril organization | 0.021289 | 4 (4) | COL5A1, HSPA8, LTBP2, COL3A1 |
| | GO:0000281 | mitotic cytokinesis | 0.021951 | 5 (5) | PLK1, KIF23, NUSAP1, RACGAP1, KIF20A |
| | GO:0007067 | mitotic nuclear division | 0.025966 | 10 (8) | PLK1, KIF23, NUSAP1, CDT1, NDC80, SMC2, KIF18B, RACGAP1 |
| | GO:0000278 | mitotic cell cycle | 0.025966 | 18 (10) | CDK1, PLK1, KIF23, CDC20, CDT1, SMC2, CCNB3, KIF18B, RACGAP1, FGFR1 |
| | GO:0061640 | cytoskeleton-dependent cytokinesis | 0.033177 | 5 (5) | PLK1, KIF23, NUSAP1, RACGAP1, KIF20A |
| | GO:1903047 | mitotic cell cycle process | 0.039601 | 15 (11) | CDK1, PLK1, KIF23, CDC20, CDT1, SMC2, CCNB3, KIF18B, RACGAP1, KIF20A, FGFR1 |
| M8SvsM4S | GO:0009157 | deoxyribonucleoside monophosphate biosynthetic process | 1 | 2 (2) | TK1, DCTD |
| | GO:0009162 | deoxyribonucleoside monophosphate metabolic process | 1 | 2 (2) | TK1, DCTD |
| | GO:0006631 | fatty acid metabolic process | 1 | 5 (5) | CPT1A, PDK4, ELOVL1, ACSL1, SREBF1 |
| | GO:0035384 | thioester biosynthetic process | 1 | 2 (2) | PDK4, ELOVL1 |
| | GO:0071616 | acyl-CoA biosynthetic process | 1 | 2 (2) | PDK4, ELOVL1 |

**Notes.**
GO, Gene Ontology; BP, Biological Process; DEGs, differentially expressed genes: the first number is the amount of all genes enriched in the BP term, and the second number is the genes listed in the next column.

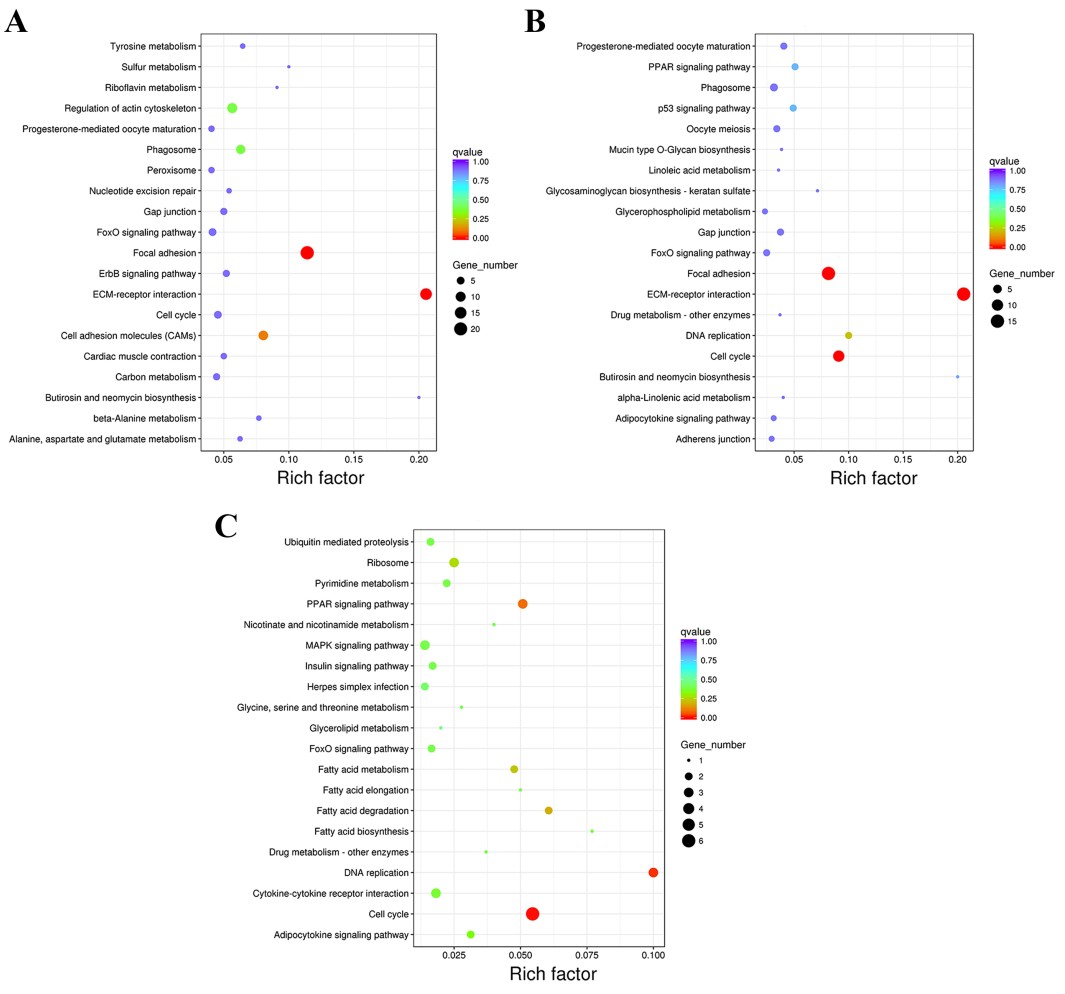

**Figure 7  The first 20 KEGG pathways of DEGs.** (A) The first 20 KEGG pathways of F8FvsF4F. (B) The first 20 KEGG pathways of F8SvsF4S. (C) The first 20 KEGG pathways of M8SvsM4S.

cytoskeleton, PPAR signaling pathway, p53 signaling pathway, insulin signaling pathway, etc. Although some of them have not reached the significant level (corrected *P*-value ≤0.05), they still have an important reference in production and theoretical research. Two, three, and two KEGG pathways were significantly enriched of the DEGs from the groups F8FvsF4F (Fig. 7A), F8SvsF4S (Fig. 7B), and M8SvsM4S (Fig. 7C), respectively. These terms include ECM–receptor interaction, focal adhesion, cell cycle, and DNA replication. Detailed information of them can be found in Table 3, and they play an important role in both the production and mechanism research of chickens.

## Verification of RNA-seq results using qPCR

We selected at least two DEGs with a fold change (FC) of ≥2 in each comparison group for verification of RNA-seq (Fig. 8). Most of them are supposed to be related to the growth of chicken. The result of qPCR and RNA-seq for the nine DEGs showed the same trend in expression, which further illustrates the accuracy of RNA-seq.

**Table 3  The KEGG pathways related to the growth and development of skeletal muscle in Jinghai yellow chicken.**

| Group | KEGG ID | KEGG pathway | Corrected p-Value | DEGs | Important genes |
|---|---|---|---|---|---|
| F8FvsF4F | gga04512 | ECM-receptor interaction | 3.95E−07 | 15 (6) | COL5A1, COLOA2, COL6A1, ITGA9, HSPG2, FN1 |
|  | gga04510 | Focal adhesion | 2.08E−06 | 21 (6) | PAK1, COL5A1, PIK3CD, COL4A2, COL2A1, COL5A2 |
| F8SvsF4S | gga04512 | ECM-receptor interaction | 1.85E−10 | 15 (6) | COL5A1, COL6A3, COL6A1, LAMC1, COL1A2, COL1A1 |
|  | gga04510 | Focal adhesion | 9.48E−06 | 15 (5) | COL5A1, COL6A3, LAMC1, LAMB1, COL1A1 |
|  | gga04110 | Cell cycle | 0.000274 | 10 (5) | CDK1, WEE1, PLK1, CDC20, CCNB3 |
| M8SvsM4S | gga04110 | Cell cycle | 0.003644 | 6 (6) | MCM2, CDC20, TGFB3, MCM5, MCM3, CCNB3 |
|  | gga03030 | DNA replication | 0.021531 | 3 (3) | MCM2, MCM3, MCM5 |

**Notes.**

KEGG, Kyoto Encyclopedia of Genes and Genomes; DEGs, differentially expressed genes: the first number is the amount of all genes enriched in the KEGG pathway, and the second number is the genes listed in the next column.

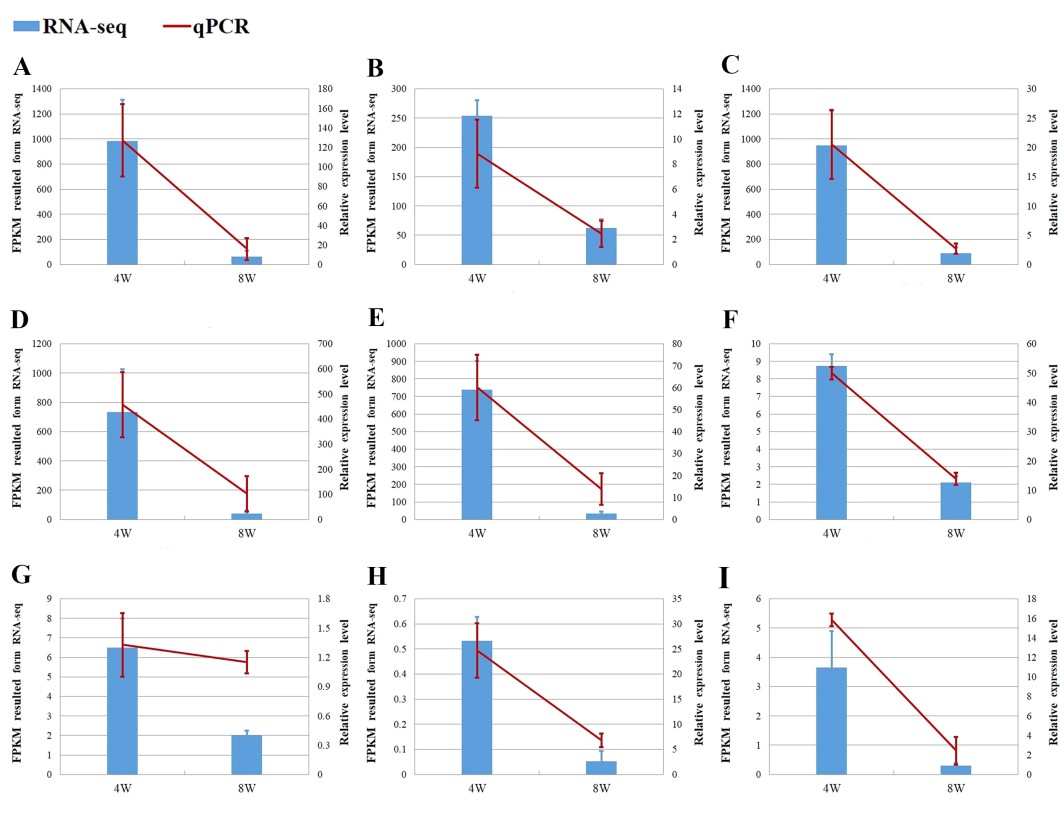

**Figure 8  The verification of RNA-seq.** (A–C) MYH1F, COL3A1 and ACTC1 from DEGs of F8SvsF4S. (D–G) MYH1F, ACTC1, C1QC and ANXA1 from DEGs of F8FvsF4F. (H, I) PLPPR4 and RACGAP1 from DEGs of M8SvsM4S.

## DISCUSSION

The skeletal muscle largely affects the yield of chicken meat in the broiler industry. The exploration of the growth and development of the skeletal muscle in broilers would help us increase chicken production. Although it has been studied by some researchers, the

specific regulation mechanisms of the skeletal muscle still remains unclear. In this study, we selected the breast muscles of Jinghai yellow chickens at growth stage for transcriptome sequencing. The results showed that 364, 219, and 111 DEGs were obtained from the three comparison groups F8FvsF4F, F8SvsF4S, and M8SvsM4S, respectively. There were three common DEGs: ADAMTS20, ARHGAP19, Novel00254, and they were all highly expressed at four weeks of age. In addition, we also found many other genes related to growth and development through analysis, such as ANXA1, ANXA2, COL1A1, COL1A2, COL2A1, and other collagen family genes; myosin heavy chain family genes (MYH15, MYH1D, MYH1F); genes of transcription factors and their receptor family members TGFB3, TGFBR3, and so on.

ADAMTS20 belongs to the ADAMTS (a disintegrin and metallopeptidase with thrombospondin motifs) proteins family, which is a secretory metalloproteinase (*Hubmacher & Apte, 2015*). ADAMTS proteins generally contain multiple domains including a metalloproteinase domain and a disintegrin domain, which could both have an important impact on the extracellular matrix (ECM) (*Rivera et al., 2019*; *Zhang et al., 2019b*; *Zhong & Khalil, 2019*). ECM participates in cell proliferation, differentiation, and migration, and then regulates chicken growth and development (*Iyoda & Fukai, 2012*; *Katayama et al., 2018*). In birds and mammals, ADAMTS20 regulates the terminal migration of melanoblasts, which eventually forms the integumental color patterns of vertebrates (*Kelsh et al., 2009*). *Wolf et al. (2015)* found that ADAMTS20 mutation also caused cleft lip and palate in dogs and humans. *Cooper et al. (2008)* found that pk-1 and pk-2 exhibited conserved synteny with ADAMTS20 and ADAMTS9 by bioinformatics analysis, and they speculated that these genes together play a regulatory role during embryogenesis and early organogenesis of chickens. In the study, we found that the ADAMTS20 expression level decreased significantly at eight weeks of age in all three comparison groups. In addition, we also found some other DEGs in this family, such as ADAMTS1 (F8SvsF4S), ADAMTS19 (F8FvsF4F), and ADAMTSL2 (F8FvsF4F), and the expression of the three genes all decreased significantly at eight weeks of age. Therefore, we infer that this family of genes plays an important regulatory role in the early growth and development of Jinghai yellow chickens.

ARHGAP19 (Rho GTPase activating protein 19) is a Rho GTPase-activating protein of the Arh GTPase-activating protein family that is involved in cell migration and actin regulation (*Lv et al., 2007*). *Amelio et al. (2012)* induced silencing of ARHGAP19, resulting in inhibition of proliferation in primary keratinocytes. *Marceaux & Petit (2018)* found that phosphorylation of ARHGAP19 by CDK1 and ROCK appeared to be essential during Kit 225 cell division, because mutation of either the CDK1 or ROCK phosphorylation sites resulted in cytokinesis failure and cell multinucleation. As a common differentially expressed gene among the three comparison groups, we found that the expression of ARHGAP19 was significantly higher at four weeks than eight weeks. At the same time, we found that another differentially expressed gene, ARHGAP12 (F8SvsF4S, M8SvsM4S), in this family also showed this expression trend. These results suggest that these family genes may play a more important role in the early growth of Jinghai yellow chickens. The third common differentially expressed gene Novel00254 was also a highly expressed gene at four

weeks of age. It may play an important role in the early development of Jinghai yellow chickens.

In addition to the above mentioned three common DEGs, we also found several DEGs related to growth and development in different groups. Both ANXA1 and ANXA2 are DEGs in the F8FvsF4F group and they were all highly expressed at four weeks of age in the study. Annexin (ANX) is a kind of protein superfamily with high abundance, calcium-dependent, and binding to negatively charged membrane phospholipids (*Rentero et al., 2018*; *Xie et al., 2018*). One of the driving forces underlying cell migration and intercellular interactions is the reorganization of cell membranes and remodeling of the cytoskeleton, processes facilitated by Annexin proteins in multiple systems (*Shah, Schiffmacher & Taneyhill, 2017*). Therefore, it can regulate many cell behaviors. *Bizzarro et al. (2010)* found that the expression of the ANXA1 gene was upregulated when a large number of mouse myoblasts C2C12 differentiated. On the contrary, when the differentiation of C2C12 was inhibited, the expression of ANXA1 was downregulated. It was speculated that the expression of the ANXA1 gene was positively correlated with the differentiation of the C2C12 myoblasts. *Leikina et al. (2015)* further used wild type and Anx A1 knockout mice as experimental materials to prove that ANXA1 could facilitate myoblast fusion in vivo. At present, no study of ANXA2 in myoblasts has been reported, but many studies have shown that ANXA2 is closely related to the proliferation and migration of various cancer cells (*Cardoso et al., 2019*; *Liu et al., 2019*; *Wei et al., 2018*). Combining these two genes with their decreased expression in our experiment, we speculate that they may also promote the fusion of chicken myoblasts and regulate muscle formation.

Most of the DEGs found in this study are members of the collagen family, including 1–6, 14, 18, and 20 types of the collagen family. Collagen is an important protein in animal connective tissue. It is also the main component of extracellular matrix (ECM) and could provide support for cell growth (*Sorushanova et al., 2019*). Osteogenesis imperfect (OI), also known as brittle bone disease, is a genetic connective tissue disorder with genetic and phenotypic diversity (*Friedrich, Scheuer & Holtje, 2019*; *Subramanian & Viswanathan, 2019*). Nearly 85% of OI patients belong to type I to type IV of OI, which is induced by COL1A1 and COL1A2 genes encoding pro α1(I) and pro α2(I) of type I procollagen in an autosomal-dominant inherited form (*Lu et al., 2019*). Type II collagen is of major importance in endochondral bone formation, growth, and normal joint function, and one of the clinical manifestations of type II collagen disease is skeletal dysplasia (*Gregersen & Savarirayan, 1993*). In addition, other collagen proteins and their coding genes (COL3A1, COL4A2, COL5A1, COL5A2, COL6A1, and so on) have also been found to be closely related to the development of osteogenic (*Egusa et al., 2007*; *Izu et al., 2016*; *Shen et al., 2018*; *Volk et al., 2014*; *Wen et al., 2019*). In the study, GO and KEGG analysis of DEGs of collagen family genes were carried out. The results showed that the family genes, especially COL1A1 and COL1A2, covered a large number of significantly enriched BP terms and KEGG pathways, which further illustrated the importance of the family genes in the early development of Jinghai yellow chicken.

In this study, three differentially expressed myosin heavy chain family genes were obtained, which are MYH1D, MYH1F, and MYH15, respectively. Myosin consists of

light (MYLC) and heavy (MYHC) chains (*Ojima, 2019*). The major myosin heavy chain subtypes expressed in tissues determine the type of muscle fibers (*Francisco et al., 2011*). Muscle fiber type is also closely related to meat quality, which determines the pH value, tenderness, intramuscular fat, and other traits related to the muscle (*Shen et al., 2015*; *Ryu & Kim, 2005*; *Zhang et al., 2014*). The experimental data of *Ahn et al. (2018)* showed that Myh1 is associated with slow muscle composition, with overexpression of Myh1 in muscle tissues being a key in modulating muscle fiber types. *Zhang et al. (2019a)* performed transcriptional analysis of the pectoral muscles of three chicken breeds (White Broiler, Daheng, and Commercial Layers of Roman) to explore the regulators mediating breast muscle growth and development, and the result showed that MYH15, as a differentially expressed gene, was highly expressed in high-weight chickens. In this experiment, MYH15 was found to be significantly highly expressed in the F4F than the F8F. These results suggest that the gene might also regulates the formation of muscle fiber types in Jinghai yellow chicken.

Another well-known family of genes related to growth and development is the transcriptional growth factor and its receptor family genes, whose members were also found in this experiment (TGFB3, TGFBR3). Many studies have shown that TGF-β3 was related to bone growth (*Haghighizadeh et al., 2019*; *He et al., 2019*; *Li et al., 2017*). Some experiments have shown that the gene also regulates skeletal muscle development. *Lu, Chen & Yang (2013)* found that TGF-β3 may play important roles during fetal myogenesis in a chicken's hindlimb. The study of *Otani et al. (2015)* showed that TGF-β3 is one of the extracellular factors regulating CTRP3 expression during myogenesis in C2C12. TGFBR3, also known as betaglycan, is the most abundantly expressed member of TGFBRs and it can bind all three TGF-β forms with high affinity (*Vander Ark, Cao & Li, 2018*). Therefore, as an important member of TGF-β superfamily signaling pathways, it plays an important role in regulating cell functions (*Gatza, Oh & Blobe, 2010*). In the study, TGFB3 and TGFBR3 were found as DEGs from the comparison groups F8SvsF4S and M8SvsM4S, respectively. Therefore, the results further support their important role in regulating the growth and development of chickens.

In addition to the DEGs discussed above, we also found some others related to chicken growth, such as WNT5B, IGFBP4, ACTC1, PAK1, SAMD11, etc. However, the regulation of genes is not independent in vivo. Genes coordinate and interact with each other to form a whole network. Therefore, we would discuss the similarities and differences between genders or groups with different growth speeds from the perspectives of GO and KEGG.

The GO enrichment analysis of DEGs in the three comparison groups (M8SvsM4S, F8FvsF4F, and F8SvsF4S) was performed. GO terms with a corrected *P*-value ≤0.05 was not found in the M8SvsM4S group, but we still found several items related to growth in the top 30 GO terms, such as deoxyribonucleoside monophosphate biosynthetic process, deoxyribonucleoside monophosphate metabolic process, and fatty acid metabolic process. They might also have an influence on the growth of slow-growing male chickens. The GO enrichment analysis of the F8FvsF4F and F8SvsF4S comparison showed that there were some terms with a corrected *P*-value ≤0.05, including 4 and 17 BP terms, respectively. Two BP terms extracellular matrix organization and extracellular structure organization

were both significantly enriched. It indicated that the two BP terms might play a more important role than others in both fast- and slow-growing groups of female chickens.

KEGG pathway analysis of DEGs in the three comparison groups was also performed and the top 20 pathways of each group are shown in Fig. 7. We found that eight pathways in the top 20 were enriched in both F8FvsF4F and the F8SvsF4S groups. But only the top two and three pathways from F8FvsF4F and the F8SvsF4S groups reached the standard with corrected $P$-value $\leq 0.05$, and the two pathways are exactly the same as two of the three. The same two pathways were ECM–receptor interaction and focal adhesion, and studies have shown that cell growth, development, and regulation were closely related to them (*Burridge, 2017*; *Thomas, Engler & Meyer, 2015*). At the same time, 58 common DEGs were also found in the two comparison groups (Fig. 2). The analysis above shows that although the growth rates of female chickens were different (a fast-growing group and a slow-growing group), the growth and development regulation mechanism of the female chickens share much in common.

However, five and two pathways in the top 20 of groups F8SvsF4S and F8FvsF4F were the same as that of the M8SvsM4S group, respectively. No common pathway with a corrected $P$-value $\leq 0.05$ was found between groups F8FvsF4F and M8SvsM4S, and only one pathway with a corrected $P$-value $\leq 0.05$ was found between groups F8SvsF4S and M8SvsM4S. We also found that there were only 7 and 18 common DEGs in the two comparison groups (Fig. 2). These results suggest that great differences in the regulation mechanisms of growth might exist between male and female chickens.

## CONCLUSION

This study examined the theoretical basis further to reveal new insights into the growth and development mechanism of chickens. Overall, the common DEGs, including ADAMTS20, ARHGAP19, Novel00254, and significantly enriched pathways, such as ECM–receptor interaction and focal adhesion, showed to be essential to the growth of chickens. Still, their functions need to be further investigated. Therefore, these results could serve as an important guide for future experimental designs of gene function in poultry science.

## ACKNOWLEDGEMENTS

The authors thank Novogene for the bioinformatics analysis.

### Funding

This study was financially supported by the Natural Science Foundation of Jiangsu Province (BK20181453), the Special Funds Project for Transforming Scientific and Technological Achievements in Jiangsu Province (BA2018099), the New Agricultural Breeds Creation Projects in Jiangsu Province (PZCZ201730), the China Agriculture Research System (CARS-41), and the Priority Academic Program Development of Jiangsu Higher Education

Institutions. The funders had no role in study design, data collection and analysis, decision to publish, or preparation of the manuscript.

## Grant Disclosures

The following grant information was disclosed by the authors:

Natural Science Foundation of Jiangsu Province: BK20181453.

Transforming Scientific and Technological Achievements in Jiangsu Province: BA2018099.

New Agricultural Breeds Creation Projects in Jiangsu Province: PZCZ201730.

China Agriculture Research System: CARS-41.

Priority Academic Program Development of Jiangsu Higher Education Institutions.

## Competing Interests

The authors declare there are no competing interests.

## Author Contributions

- Pengfei Wu conceived and designed the experiments, performed the experiments, analyzed the data, prepared figures and/or tables, authored or reviewed drafts of the paper, and approved the final draft.
- Xinchao Zhang, Fuxiang Chen and Mingliang He performed the experiments, prepared figures and/or tables, and approved the final draft.
- Genxi Zhang conceived and designed the experiments, analyzed the data, prepared figures and/or tables, authored or reviewed drafts of the paper, and approved the final draft.
- Tao Zhang conceived and designed the experiments, analyzed the data, authored or reviewed drafts of the paper, and approved the final draft.
- Jinyu Wang analyzed the data, authored or reviewed drafts of the paper, and approved the final draft.
- Kaizhou Xie performed the experiments, authored or reviewed drafts of the paper, and approved the final draft.
- Guojun Dai conceived and designed the experiments, authored or reviewed drafts of the paper, and approved the final draft.

## Animal Ethics

The following information was supplied relating to ethical approvals (i.e., approving body and any reference numbers):

The Animal Ethics Committee of Yangzhou University approved the study (Yzu DWLL-201903-001).

## Data Availability

Data is available at the NCBI Sequence Read Archive using accession numbers: SAMN12173007 (F4F_1), SAMN12173008 (F4F_2), SAMN12173009 (F4F_3), SAMN12173019 (F8F_1), SAMN12173020 (F8F_2), SAMN12173021 (F8F_3), SAMN12173010 (F4S_1), SAMN12173011 (F4S_2), SAMN12173012 (F4S_3), SAMN12173022 (F8S_1), SAMN12173023 (F8S_2), SAMN12173024 (F8S_3),

SAMN12173016 (M4S_1), SAMN12173017 (M4S_2), SAMN12173018 (M4S_3), SAMN12173028 (M8S_1), SAMN12173029 (M8S_2), SAMN12173030 (M8S_3).

## Supplemental Information

Supplemental information for this article can be found online at http://dx.doi.org/10.7717/peerj.8950#supplemental-information.

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

# PeerJ

**Papah MB, Brannick EM, Schmidt CJ, Abasht B. 2018.** Gene expression profiling of the early pathogenesis of wooden breast disease in commercial broiler chickens using RNA-sequencing. *PLOS ONE* **13(12)**:e0207346 DOI 10.1371/journal.pone.0207346.

**Prucha M, Zazula R, Russwurm S. 2018.** Sepsis diagnostics in the era of omics technologies. *Prague Medical Report* **119(1)**:9–29 DOI 10.14712/23362936.2018.2.

**Ren T, Li Z, Zhou Y, Liu X, Han R, Wang Y, Yan F, Sun G, Li H, Kang X. 2018.** Sequencing and characterization of lncRNAs in the breast muscle of Gushi and Arbor Acres chickens. *Genome* **61(5)**:337–347 DOI 10.1139/gen-2017-0114.

**Rentero C, Blanco-Munoz P, Meneses-Salas E, Grewal T, Enrich C. 2018.** Annexins-coordinators of cholesterol homeostasis in endocytic pathways. *International Journal of Molecular Science* **19(5)**:1444 DOI 10.3390/ijms19051444.

**Rivera S, Garcia-Gonzalez L, Khrestchatisky M, Baranger K. 2019.** Metalloproteinases and their tissue inhibitors in Alzheimer's disease and other neurodegenerative disorders. *Cellular and Molecular Life Sciences* **76(16)**:3167–3191 DOI 10.1007/s00018-019-03178-2.

**Roelfsema F, Yang RJ, Takahashi PY, Erickson D, Bowers CY, Veldhuis JD. 2018.** Effects of Toremifene, a selective estrogen receptor modulator, on spontaneous and stimulated GH secretion, IGF-I, and IGF-binding proteins in healthy elderly subjects. *Journal of Endocrine Society* **2(2)**:154–165 DOI 10.1210/js.2017-00457.

**Ryu YC, Kim BC. 2005.** The relationship between muscle fiber characteristics, post-mortem metabolic rate, and meat quality of pig longissimus dorsi muscle. *Meat Science* **71(2)**:351–357 DOI 10.1016/j.meatsci.2005.04.015.

**Shah A, Schiffmacher AT, Taneyhill LA. 2017.** Annexin A6 controls neuronal membrane dynamics throughout chick cranial sensory gangliogenesis. *Developmental Biology* **425(1)**:85–99 DOI 10.1016/j.ydbio.2017.03.011.

**Shen LY, Luo J, Lei HG, Jiang YZ, Bai L, Li MZ, Tang GQ, Li XW, Zhang SH, Zhu L. 2015.** Effects of muscle fiber type on glycolytic potential and meat quality traits in different Tibetan pig muscles and their association with glycolysis-related gene expression. *Genetics and Molecular Research* **14(4)**:14366–14378 DOI 10.4238/2015.November.13.22.

**Shen Y, Jing D, Hao J, Tang G, Yang P, Zhao Z. 2018.** The effect of beta-aminopropionitrile on skeletal micromorphology and osteogenesis. *Calcified Tissue International* **103(4)**:411–421 DOI 10.1007/s00223-018-0430-4.

**Sorushanova A, Delgado LM, Wu Z, Shologu N, Kshirsagar A, Raghunath R, Mullen AM, Bayon Y, Pandit A, Raghunath M, Zeugolis DI. 2019.** The collagen suprafamily: from biosynthesis to advanced biomaterial development. *Advanced Materials* **31(1)**:e1801651 DOI 10.1002/adma.201801651.

**Subramanian S, Viswanathan VK. 2019.** *Osteogenesis Imperfecta.* StatPearls. Treasure Island (FL): StatPearls Publishing LLC.

**Thomas K, Engler AJ, Meyer GA. 2015.** Extracellular matrix regulation in the muscle satellite cell niche. *Connective Tissue Research* **56(1)**:1–8 DOI 10.3109/03008207.2014.947369.

**Trapnell C, Williams BA, Pertea G, Mortazavi A, Kwan G, Van Baren MJ, Salzberg SL, Wold BJ, Pachter L. 2010.** Transcript assembly and quantification by RNA-Seq reveals unannotated transcripts and isoform switching during cell differentiation. *Nature Biotechnology* **28(5)**:511–515 DOI 10.1038/nbt.1621.

**Vander Ark A, Cao J, Li X. 2018.** TGF-beta receptors: in and beyond TGF-beta signaling. *Cellular Signalling* **52**:112–120 DOI 10.1016/j.cellsig.2018.09.002.

**Volk SW, Shah SR, Cohen AJ, Wang Y, Brisson BK, Vogel LK, Hankenson KD, Adams SL. 2014.** Type III collagen regulates osteoblastogenesis and the quantity of trabecular bone. *Calcified Tissue International* **94(6)**:621–631 DOI 10.1007/s00223-014-9843-x.

**Wang Z, Gerstein M, Snyder M. 2009.** RNA-Seq: a revolutionary tool for transcriptomics. *Nature Reviews Genetics* **10(1)**:57–63 DOI 10.1038/nrg2484.

**Wei DM, Dang YW, Feng ZB, Liang L, Zhang L, Tang RX, Chen ZM, Yu Q, Wei YC, Luo DZ, Chen G. 2018.** Biological effect and mechanism of the miR-23b-3p/ANXA2 axis in pancreatic ductal adenocarcinoma. *Cellular Physiology and Biochemistry* **50(3)**:823–840 DOI 10.1159/000494468.

**Wen Y, Yang H, Wu J, Wang A, Chen X, Hu S, Zhang Y, Bai D, Jin Z. 2019.** COL4A2 in the tissue-specific extracellular matrix plays important role on osteogenic differentiation of periodontal ligament stem cells. *Theranostics* **9(15)**:4265–4286 DOI 10.7150/thno.35914.

**Wolf ZT, Brand HA, Shaffer JR, Leslie EJ, Arzi B, Willet CE, Cox TC, McHenry T, Narayan N, Feingold E, Wang X, Sliskovic S, Karmi N, Safra N, Sanchez C, Deleyiannis FW, Murray JC, Wade CM, Marazita ML, Bannasch DL. 2015.** Genome-wide association studies in dogs and humans identify ADAMTS20 as a risk variant for cleft lip and palate. *PLOS Genetics* **11(3)**:e1005059 DOI 10.1371/journal.pgen.1005059.

**Xie XL, Yang H, Chen LN, Wei Y, Zhang SH. 2018.** ANXC7 Is a mitochondrion-localized annexin involved in controlling conidium development and oxidative resistance in the thermophilic fungus thermomyces lanuginosus. *Frontiers in Microbiology* **9**:1770 DOI 10.3389/fmicb.2018.01770.

**Young MD, Wakefield MJ, Smyth GK, Oshlack A. 2010.** Gene ontology analysis for RNA-seq: accounting for selection bias. *Genome Biology* **11(2)**:R14 DOI 10.1186/gb-2010-11-2-r14.

**Zhang Z, Du H, Yang C, Li Q, Qiu M, Song X, Yu C, Jiang X, Liu L, Hu C, Xia B, Xiong X, Yang L, Peng H, Jiang X. 2019a.** Comparative transcriptome analysis reveals regulators mediating breast muscle growth and development in three chicken breeds. *Animal Biotechnology* **30**:1–9 DOI 10.1080/10495398.2018.1476377.

**Zhang M, Liu YL, Fu CY, Wang J, Chen SY, Yao J, Lai SJ. 2014.** Expression of MyHC genes, composition of muscle fiber type and their association with intramuscular fat, tenderness in skeletal muscle of Simmental hybrids. *Molecular Biology Reports* **41(2)**:833–840 DOI 10.1007/s11033-013-2923-6.

**Zhang Y, Wang F, Chen G, He R, Yang L. 2019b.** LncRNA MALAT1 promotes osteoarthritis by modulating miR-150-5p/AKT3 axis. *Cell & Bioscience* **9**:54 DOI 10.1186/s13578-019-0302-2.

**Zhong S, Khalil RA. 2019.** A disintegrin and metalloproteinase (ADAM) and ADAM with thrombospondin motifs (ADAMTS) family in vascular biology and disease. *Biochemical Pharmacology* **164**:188–204 DOI 10.1016/j.bcp.2019.03.033.

**Zhu X, Li YL, Liu L, Wang JH, Li HH, Wu P, Chu WY, Zhang JS. 2016.** Molecular characterization of Myf5 and comparative expression patterns of myogenic regulatory factors in Siniperca chuatsi. *Gene Expression Patterns* **20(1)**:1–10 DOI 10.1016/j.gep.2015.10.003.