# Peer review of "Transcriptome for the breast muscle of Jinghai yellow chicken at early growth stages"

_PeerJ, doi:10.7717/peerj.8950_

## Round 0.1 · original submission · Major Revisions

Dear authors,

I have received comments on your paper. I give you a chance to carefully improve the quality of your paper. I also recommend a professional English editing of your paper.

The statistical analysis section needs careful revision as well as the presentation of the tables and figures.

Kind regards

Reviewer 1 ·

Basic reporting

Major issues
- Throughout the manuscript there is an issue with grammar and written English which is very poor in parts - it should be edited by a native English speaker before being re-submitted.
- The discussion section starts nice, with good appointments, but at some point, I think you're lost. And, the final part is a little bit confusing. Re-write the last paragraphs of this section and give better closure.

Minor issues
Some of the suggested corrections are in the .docx file (attached).

Lines 50-52: The background description is poor. Write again, please.
Lines 73-80: Re-write this paragraph with better grammar.
Line 86: change this expression - "biggest factor".
Lines 94-95: Re-write this sentence with better grammar - "Among them, transcriptome...".
Lines 104-104: Split this sentence to better comprehension.
Line 119: explain "lnc" from lncRNAs. All abbreviations should be explained in parenthesis when it appeared for the first time.
Lines 129-134: Let this information for the Materials and Methods section.
Lines 219-221: Re-write this sentence as it doesn't make sense as is.
Lines 236-237: change to "Figure 1 shows the DEG with P-adjust ≤ 0.05".
Line 237: Do not start a sentence with numerals.
Line 240: Do not start a sentence with numerals.
Line 243: Do not start a sentence with numerals.
Line 246: Do not start a sentence with numerals.
Line 250: The Ensemble gene ID is not necessary, add gene description inside the parenthesis.
Line 251: change the word "strains".
Lines 273-275: Re-write this sentence.
Line 286: Change the expression "by many scholars".
Line 403: change the expression "important significance".
Lines 408-409: Re-write this sentence.

Tables
Table 1: change "was very significant" to "was considered as significant".

Table 2: Tables should be self-explanatory, please provide more information in the title (ex. BP = biological process). I found it very vague to present "Some genes related to growth and development". Choose a fixed number of genes to show, or present all enriched genes for that BP.

Experimental design

- Let the goals of your study clearer in the introduction.

Validity of the findings

- The conclusion is weak. You could name some interesting genes that could be investigated in future studies.

Additional comments

I think the work was well done, but some issues have to be clarified. Please review the written English and grammar to improve your paper.

Annotated reviews are not available for download in order to protect the identity of reviewers who chose to remain anonymous.

Reviewer 2 ·

Basic reporting

Please review in whole document the way to cite.

Abstract must contain: Background, Methods, Results, Conclusion and key words, this paper hahas not include Colclusion in the Abstract.

Please mention in material and methods that the experiment was approved by ethics Committee.

Please check the lines
152 (poly-N)
338- pH value

Experimental design

That kind of research shed light about genes expression in different kinds of conditions.

They are relevant for future experiment designs and can be guided to propose experimental design on gene function.

The experiment ordering sequential and the results were well interpreted.

Methods were sufficiently described, sequential and corresponding to the aim of the work.
Authors should mention the elapsed time between taking the samples and RNA extraction process.

Software used to stimated gene expression is okey, as well the to use FPKM to normalizated estimation of gene expression.

Validity of the findings

Please mention criteria used to choose the genes that were verified by qPCR, they were nine (9). There is no mention on why they were selected.

It is important to show the different results on pathway between female and male, nothing is mentions on the text.

Discussion lines 253 to 255 should be improved, the paragraphs are non-sequencial, they contain many ideas but cutting the point.

Nothing is mentioned about the gen Novel 00254

Additional comments

Genes regulations is one most fascinating event in genetics, therefore any effort to understand the nature of gene regulations is invaluable. This paper is important step to know gene expression and regulation in an important economic specie.

---

## Round 0.2 · Minor Revisions

Dear authors,

Please, I invite your to revise your paper according to the last reviewer who is happy with what you did but suggests more modifications to improve the quality of your manuscript.

Kind regards

Reviewer 1 ·

Basic reporting

I consider that this paper has been well revised, and the writing is more fluid and pleasant. However, I still have some suggestions to improve the paper.

Suggestions:

Lines 59-60: change the sentence for "The differential gene expression analysis resulted in 364, 219 and 111 DEGs (P-value < 0.05) for the three comparison groups, F8FvsF4F, F8SvsF4S, and M8SvsM4S, respectively."
I believe that this way, the sentence becomes clearer.

Line 63: change the sentence for "such as ANXA1, COL1A1, MYH15, TGFB3 and, ACTC1, were obtained.".

Line 64: change "(58)" for "(n = 58)". Just the number in the parenthesis seems like a reference.

Line 95: change "transcriptome" to "transcriptomics".

Line 96: change "genomics" to "omics" - just a suggestion.

Line 97: change "transcriptome" to "transcriptomic".

Lines 98-101: The sentence is too long and with many repeated words, becoming even a little redundant. I suggest that you summarize this concept a little, making the sentence more direct.

Line 102: change "has become one of the important means" to "has become an important tool".

Line 106: remove "were".

Lines 120-121: change "in muscle tissues, TCONS_00064133 and TCONS_00069348, were identified." to " in muscle tissues, the TCONS_00064133 and the TCONS_00069348.".

Lines 121-123: suggestion, change the last sentence for "Although RNA-seq technology has been applied to study the growth and development of poultry, the specific regulatory mechanisms of skeletal muscle development remain unclear, and the transcriptome sequencing technology still will be increasingly explored in future studies in the field.".

Line 152: Insert a comma after the parenthesis.

Line 191: change "RNA" to "RNA samples".

Line 221-223: suggestion, change the sentence to "Figure 1 shows the DEG with adjusted P-values of ≤0.05. The analysis resulted in 364, 219, and 111 DEGs obtained from the three comparison groups F8FvsF4F (Fig 1A ), F8SvsF4S (Fig 1B), and M8SvsM4S (Fig 1C), respectively."

Line 225-226: suggestion, change the sentence to "And, 27 upregulated genes and 84 downregulated genes in the M8SvsM4S group.".

Line 227: change "of" to "with".

Line 232: insert "groups" after M8SvcM4S.

Line 238-240: change the sentences "The clustering results are shown in Figures 3–5. The figures demonstrate that the three individuals in each group were all well clustered together. The results show that the three individuals selected in each group have good repeatability." to "The clustering results are shown in Figures 3–5, demonstrating that the three individuals in each group were all well clustered together. Also, these results show that the three individuals selected in each group have good repeatability.".

Line 243: remove "the", start with "Significantly enriched GO terms...".

Line 244-246: change the sentence to "For the F8FvsF4F (Fig 6A) and F8SvsF4S (Fig 6B) groups, 12 and 44 GO terms were significantly enriched, within these 4 and 17 were classified as biological processes (BP), respectively. ".

Line 246-247: change the sentence to "In Table 2, we highlighted some important BP terms related to growth and development, such as extracellular matrix organization, extracellular structure organization, cell adhesion, cell division, and fibril organization.".

Line 249: insert a comma before "and".

Line 250: exclude "respectively".

Line 310: insert "mentioned" after "above".

Line 393-394: change "We could find that eight pathways in the top 20 were enriched in both, the F8FvsF4F group and the F8SvsF4S group" to We found that eight pathways in
top 20 were enriched in both F8FvsF4F and F8SvsF4S groups".

Line 395-396: the sentence is a little bit confusing. I suggest re-write it.

Line 410-413: it should be in the conclusion section.

Line 415-418: Conclusion, my suggestion is to change to "This study examined the theoretical basis further to reveal new insights into the growth and development mechanism of chickens. Overall, the common DEGs, including ADAMTS20, ARHGAP19, Novel00254, and significantly enriched pathways, such as ECM–receptor interaction and focal adhesion, showed to be essential to the growth of chickens. Still, their functions need to be further investigated. Therefore, these results could serve as an important guide for future experimental designs of gene function in poultry science."

**I highlighted all the sentences above in the PDF file attached.

Tables and figures:
- Figures description is now ok.
- Tables: I still think that you could write more robust tittles. Like: "Table 3 - The KEGG pathway related to the growth and development of skeletal muscle in

Experimental design

The experimental design is ok and, all my issues have been attended in the revised paper.

Validity of the findings

The authors discussed interesting finds for chicken muscular development and growth. These finds will be crucial for future studies in poultry science.

Additional comments

I consider that the authors have done an excellent job of proofreading. The new version of the paper is much better written and pleasant to read. I believe that all of my previous comments have been answered. However, I still have some suggestions that I think will make the work even better.

Annotated reviews are not available for download in order to protect the identity of reviewers who chose to remain anonymous.

---

## Round 0.3 · accepted · Accept

Dear authors,

I thank you for the revision of you manuscript according to the last comments from the referees. I am glad to announce the acceptance of your paper for publication in PeerJ.

Kind regards
Dr. Gagaoua